# Cell Migration in Endometriosis Responds to Omentum-Derived Molecular Cues Similar to Ovarian Cancer

**DOI:** 10.3390/ijms26051822

**Published:** 2025-02-20

**Authors:** Kah Yee Goh, Su Chin Tham, Terence You De Cheng, Ravichandran Nadarajah, Ronald Chin Hong Goh, Shing Lih Wong, Tew Hong Ho, Ghee Kheng Chew, Andy Wei Keat Tan, Hemashree Rajesh, Hong Liang Chua, Tze Tein Yong, Su Ling Yu, Jia Min Kang, Kah Weng Lau, Amos Zhi En Tay, Sangeeta Mantoo, Inny Busmanis, Sung Hock Chew, Timothy Yong Kuei Lim, Wai Loong Wong, Qiu Ju Ng, Junjie Wang, Sun Kuie Tay, Chit Fang Cheok, Darren Wan-Teck Lim, Elaine Hsuen Lim

**Affiliations:** 1Division of Medical Oncology, National Cancer Centre Singapore, 30 Hospital Boulevard, Singapore 168583, Singapore; 2Institute of Molecular and Cell Biology, A*STAR, 61 Biopolis Drive, Proteos, Singapore 138673, Singapore; 3Department of Obstetrics & Gynaecology, Singapore General Hospital, Outram Road, Singapore 169608, Singapore; 4Department of Anatomical Pathology, Singapore General Hospital, Academia, College Road, Singapore 169856, Singapore; 5Department of Gynaecological Oncology, KK Women’s and Children’s Hospital, 100 Bukit Timah Road, Singapore 229899, Singapore; 6Department of Pathology, National University Hospital, 5 Lower Kent Ridge Road, Singapore 119074, Singapore; 7Department of Pathology and Laboratory Medicine, KK Women’s and Children’s Hospital, 100 Bukit Timah Road, Singapore 229899, Singapore; 8Department of Pathology, Yong Loo Lin School of Medicine, National University of Singapore, 5 Lower Kent Ridge Road, Singapore 119074, Singapore; 9Office of Academic and Clinical Development, Duke-NUS Medical School, 8 College Road, Singapore 169857, Singapore

**Keywords:** omentum, endometriosis, ovarian cancer, migration, HGF, c-MET, PTTG1

## Abstract

Endometriosis is common and poses significant morbidity of lasting impact to young, pre-menopausal women, while ovarian cancer is a lethal gynecologic condition. Both conditions need better treatment. The human omentum is an apron of adipose tissue in the abdominopelvic cavity, the same space in which endometriosis and ovarian cancer manifest. We aim to determine molecular cues emitted by the omentum that aid the trans-coelomic spread of endometriosis and ovarian cancer in the abdomen–pelvic/peritoneal space. Endometriosis and ovarian cancer patients were prospectively recruited. Primary cell cultures of surgically-resected omentum, endometriosis and ovarian cancer were generated, and conditioned media (CM) from the omentum was derived. They were used for in vitro assays to evaluate the effect of the omentum on cell migration, angiogenesis and proliferation in endometriosis and ovarian cancer. Omental CM promoted cell migration in primary cultures of endometriosis and ovarian cancer. Omental CM contained high levels of HGF, SDF-1a, MCP-1, VEGF-A, IL-6 and IL-8. The observed cell migration was blocked by c-MET inhibition, suggesting that HGF/c-MET signaling mediates cell migration in endometriosis and ovarian cancer. Furthermore, PTTG1 was consistently upregulated in the migrated cells in both endometriosis and ovarian cancer. The omentum provides a favorable environment for trans-coelomic spread of endometriosis and ovarian cancer. HGF, c-MET and PTTG1 are potential therapeutic targets for inhibiting the abdomen–pelvic/peritoneal spread of endometriosis and ovarian cancer.

## 1. Introduction

Endometriosis is a gynecologic disorder where endometrial cells grow outside of the uterus, causing severe pelvic pain and infertility. According to the World Health Organization, endometriosis affects approximately 10% of women in their reproductive years [1]; most are asymptomatic and do not seek medical attention. Endometriosis is a benign condition, but is associated with a 2- to 3-fold increased risk of developing ovarian cancer [2,3,4,5,6,7,8,9,10]. The distribution of endometriotic lesions mimics ovarian cancer metastasis; oncogenic mutations in *ARID1A*, *PIK3CA*, *KRAS* and *PPP2R1A* have been described in endometriotic tissue [11]. These data suggest that endometriosis is less than benign, begging the question of the role of endometriosis in ovarian cancer: precursor, facilitator or other? In 1927, Sampson postulated retrograde menstruation as endometriosis’ etiology [12]. While this theory may explain the presence of endometrial tissue outside of the uterus, it does not explain its distribution pattern nor mode of spread.

Ovarian cancer is a lethal gynecologic malignancy [13,14]. A distinguishing feature of ovarian cancer is its mode of spread. Unlike other malignancies that tend to disseminate via the blood and lymphatic systems, ovarian cancer’s trans-coelomic metastasis results in lesions appearing on the peritoneal and serosal surfaces of abdominopelvic structures, including the omentum [15,16,17]. The similar distribution pattern between endometriosis and ovarian cancer led us to posit similar biological processes underlying cell migration in both conditions.

Histologically, endometriosis lesions comprise stroma and normal endometrial glands. In contrast, ovarian cancer is heterogeneous, consisting of different histological subtypes: serous, endometrioid, clear cell and mucinous carcinoma. They are thought to originate from various parts of the gynecologic anatomy, e.g., fallopian tube, transitional cell nests at the tubal–mesothelial junction and importantly, the uterine endometrium [18], the same location where endometriosis is thought to originate.

The human omentum is an apron of fatty tissue that drapes over abdominopelvic structures; its physiological role is not well-defined. It mainly consists of adipocytes and other stromal and immune cells. Prior studies have shown that the omentum functions in immune regulation, particularly in peritoneal defence, through recruitment of white blood cells to the peritoneal cavity, rapid clearance of foreign particles and formation of dense adhesions to seal off contaminants [19]. Furthermore, due to its intrinsic angiogenic properties, the omentum is a frequent site of metastasis in many malignancies, particularly ovarian cancer [19]. Specifically, adipocytes from the human omentum secrete soluble factors (e.g., IL-8, MCP-1) that might induce cell migration in ovarian cancer [20,21]. We hypothesized that cell migration in endometriosis would be in response to such molecular cues.

Here, we examine the role of human omental adipocytes in cell migration in endometriosis, comparing with ovarian cancer. Specifically, we aimed to identify factors that may exert a tropic effect on endometriosis in a paracrine fashion. Primary cell cultures were used in our experiments to maximally mimic the human physiological status, by preserving the distinction in biological profiles between benign endometriosis and malignant ovarian cancer as much as possible.

## 2. Results

### 2.1. Developing Primary Cell Cultures and Conditioned Media from Fresh Human Tissue Samples of Endometriosis, Ovarian Cancer and the Omentum

A total of 88 patients consented to study participation, where 17 were excluded as they had non-epithelial or mixed epithelial ovarian cancer, serous tubal intraepithelial carcinoma (STIC) or primary cancers of other origins (Appendix A). We included 71 patients of whom fifteen had endometriosis (EMS), eleven had benign tumors, three had borderline mucinous ovarian tumors (BorMT) and forty-two had malignant epithelial ovarian cancers of different histological subtypes: high grade serous carcinoma (HGSC), clear cell carcinoma (CCC), endometrioid carcinoma (EC) and mucinous carcinoma (MC).

We collected 71 tumors and 45 paired omenta and developed a bedside-to-bench workflow from tissue acquisition to the derivation of primary cell cultures, omental-adipose stromal cells (O-ASC) and conditioned media (CM) (Figure 1). We successfully established primary cell cultures from 39 tumors and primary O-ASC cultures from 42 omenta (Appendix A). Of these, we assayed 26 tumors and 29 omenta (Appendix A).

### 2.2. Omental-Adipose Stromal Cells Secrete Factors That Promote Cell Migration in Primary Cultures of Endometriosis and Ovarian Cancer

Using the transwell cell migration system (Figure 2A), we evaluated the effect of O-ASC CM on the migration abilities of primary cultures of endometriosis and ovarian cancer. Cell migration was assessed by fold change in the number of cells that migrated in the presence of O-ASC CM versus control medium (Appendix A). For EMS, cells were assayed with O-ASC CM from different EMS patients’ omenta, as the same patient’s corresponding omentum was unavailable. O-ASC CM from EMS patients significantly induced migration of EMS cells 1.3-, 1.6- and 1.2-fold, respectively (KO48, E128, E402; Figure 2B; Appendix A). When EMS cells were assayed with O-ASC CM from ovarian cancer, cell migration was increased more robustly at 2.3- and 2.2-fold, respectively (Figure 2C; Appendix A).

For HGSC, CCC and EC, each cell migration assay was tested with each patient’s corresponding O-ASC CM. For MC, O-ASC CM from BorMT was used, as omentum from MC was unavailable. By histological subtype, migration was increased 1.5-, 3.7- and 2.2-fold in HGSC, respectively (KO20, E268, E335; Figure 2D; Appendix A); increased 2.5-fold in CCC (E243; Figure 2E; Appendix A); increased 1.6-, 2.1- and 2.6-fold in EC, respectively (E255, E262, E337; Figure 2F; Appendix A) and increased 1.5- and 1.8-fold in MC, respectively (E127 and E261; Figure 2G; Appendix A). Essentially, the most robust responses were observed in O-ASC CM from ovarian cancer patients; O-ASC CM from EMS and BorMT cases typically induced a weaker response.

The wound healing assay further confirmed that O-ASC CM from EMS and ovarian cancer promotes migration of EMS and ovarian cancer (Appendix A). Relative to control, rate of wound closure was increased in EMS (KO48 and E402) after incubation with O-ASC CM from EMS patients. Similarly, wound closure in ovarian cancer, namely HGSC (KO20, E268, E335) and EC (E255, E262, E337), was accelerated following incubation with the respective patient’s O-ASC CM. This was also observed in MC (E261) following incubation with O-ASC CM from a BorMT case.

### 2.3. Characterizing the Secretome of Omental-Adipose Stromal Cells

To determine the factors associated with cell migration, the Luminex platform was used to simultaneously quantify multiple analytes in O-ASC CM from 29 omenta (Appendix A), comprising conditions spanning frank malignant (11 HGSC, 8 CCC, 3 EC, 1 MC) to borderline tumors (2 BorMT), endometriosis (3 EMS) and benign cyst (1 BeCy). Each O-ASC CM was interrogated for 65 cytokines. Compared to control medium, 10 cytokines were present in higher concentrations in O-ASC CM, based on the average expression values (Figure 3 and Appendix A, Appendix A). They include hepatocyte growth factor (HGF), stromal cell-derived factor 1 (SDF-1a), monocyte chemoattractant protein-1/chemokine (CC-motif) ligand 2 (MCP-1/CCL2), vascular endothelial growth factor A (VEGF-A), interleukin-6 (IL-6), interleukin-8 (IL-8), epithelial neutrophil-activating protein 78/CXC chemokine ligand 5 (ENA-78/CXCL5), growth-related oncogene-alpha/CXC chemokine ligand 1 (GROα/CXCL1), fibroblast growth factor 2 (FGF-2) and matrix metalloproteinase 1 (MMP-1). These are multi-functional proteins that have been implicated in various cancer processes, e.g., cell migration, proliferation, invasion and metastasis, as well as angiogenesis and immune response.

Within each histological type, the measured levels of each cytokine was heterogenous, but we observed certain trends. HGF, SDF-1a and MCP-1 stood out as they were consistently more highly secreted in HGSC, compared to other histological types (Figure 3A–C and Appendix A, Appendix A). There were differences in the O-ASC CM cytokine profiles between benign (EMS/BeCy) and malignant (ovarian cancer: HGSC/CCC/EC/MC) disease. The average VEGF-A and IL-6 levels were lower in EMS and BeCy, compared to ovarian cancer (Figure 3D,E, Appendix A). In contrast, the average IL-8 levels were higher in benign disease and borderline mucinous tumors, compared to ovarian cancer (Figure 3F, Appendix A).

### 2.4. Differential Gene Expression Profiles of Migrated Versus Unmigrated Cells in Ovarian Cancer and Endometriosis

To identify the genes associated with cell migration in response to O-ASC CM, we compared gene expression profiles of migrated cells versus unmigrated cells in EMS and ovarian cancer (HGSC, EC). Using the transwell cell migration system, EMS (KO48, E128, E402), HGSC (KO20, E164, E208 and E268) and EC (E255, E262, E337) were each exposed to histologically matched O-ASC CM (E377 for EMS, KO20 for HGSC, E337 for EC). Upon separation, the respective gene expression profiles of migrated and unmigrated cells were analyzed on the Nanostring nCounter platform, using the PanCancer Progression panel (Figure 4A; Appendix A).

In EMS, *PTTG1* and *ARHGDIB* were upregulated in migrated cells (Figure 4B,C; Appendix A). In HGSC, *c-MET*, *PTTG1*, *RBL1* and *CNN1* were upregulated in migrated cells, while *HDAC5* was downregulated (Figure 4D,E; Appendix A). In EC, *MAPKAPK3*, *NOTCH1*, *AGGF1*, *ID1*, *EPHA2*, *NAA15* and *PTTG1* were upregulated in migrated cells, while *HDHD3* was downregulated (Figure 4F,G; Appendix A). Notably, *PTTG1* (Pituitary Tumor Transforming Gene 1) was upregulated in the migrated cell populations of all three histological types. The upregulation of *c-MET* in the HGSC migrated population, together with an elevation of HGF in the O-ASC CM from HGSC (Figure 3A and Appendix A, Appendix A), corroborates HGF/c-MET signaling in cell migration [22,23,24,25].

### 2.5. HGF/c-MET Signaling Promotes Cell Migration in Endometriosis

To determine if HGF/c-MET signaling mediated cell migration in endometriosis and non-HGSC ovarian cancer, we first assessed *c-MET* expression level using qRT-PCR. We tested 25 tissues comprising seven EMS, one BorMT, one BeCy, nine HGSC, two MC, two CC and three EC. All showed moderate levels of *c-MET* expression with Ct values ranging 22–28 (Appendix A).

Using the transwell cell migration system, the addition of PHA665752 (small molecule c-MET inhibitor) significantly reduced the cell migration effect of O-ASC CM on EMS (KO48, E128; Figure 5A,B; Appendix A), HGSC (KO20, E268, E335), EC (E255, E262, E337) and MC (E261) (Figure 5C–I; Appendix A). These results showed that HGF/c-MET signaling promotes cell migration in endometriosis and ovarian cancer. Furthermore, HGF derived from the omentum (O-ASC CM) is functionally active and inhibitable by PHA665752.

### 2.6. O-ASC CM Derived from Endometriosis and Ovarian Cancer Cells Induce Angiogenesis but Not Cell Proliferation

Using HUVECs (human umbilical vein endothelial cells), we examined the angiogenic potential of 22 O-ASC CM, comprising three EMS (E242, E377, E402), two BorMT (E239, E329), one BeCy (E247), six HGSC (KO20, KO24, E226, E268, E335, E343), six CCC (E231, E233, E240, E243, E404, E406), three EC (E255, E262, E337) and one MC (E362) (Appendix A). The angiogenic activity was evaluated based on the number of master junctions, meshes and total tube length. Relative to control media, O-ASC CM from HGSC, MC and EMS induced significant angiogenic activity (Appendix A). Relative to EMS, HGSC and MC induced significantly stronger angiogenic activity, whereas CCC and EC had a significantly weaker effect; there was no significant difference when compared with BorMT and BeCy (Appendix A).

The MTT assay was used to evaluate the effect of O-ASC CM on cell viability in primary cultures of endometriosis and ovarian cancer (Figure 6). As the MTT assay measures the number of metabolically active cells, it is used as a proxy for cell proliferation since a higher number of viable cells reflects an increase in cell growth. In our study, we evaluated cell viability over a 6-day period because the primary cells grew at a slower rate and usually reached near-confluence on the sixth day. The O-ASC CM used was from the same patient or from the same histological type. Relative to control, O-ASC CM did not induce a significant change in cell proliferation of HGSC (KO20, E164; Figure 6A,B) nor EMS (E128, E271; Figure 6C,D). Cell proliferation was also not affected in EMS (E128, E195, KO48; Figure 6E,G) when exposed to O-ASC CM from BorMT or HGSC.

## 3. Discussion

This study compared the human omentum’s role in inducing intra-abdominal cell migration in endometriosis and ovarian cancer. The factors that led us to investigate the omentum were (i) its physical location, (ii) the presence of endometriosis or ovarian cancer on it and (iii) secretion of factors by adipocytes, the omentum’s predominant cell type, that promote carcinogenesis [26]. We examined primary cultures established from tissue samples of endometriosis, ovarian cancer and the omentum to maximally reflect their true physiological states, in contrast with using established cell lines that already harbor genomic aberrations that confer cancer characteristics, e.g., immortality. Using primary cultures more accurately represents the tissue’s stromal cell subpopulations, which may help to sustain tissue survival ex vivo in the laboratory.

Our findings showed that the human omentum induces cell migration in endometriosis, similar to the tropism displayed by ovarian cancer for the omentum [27,28,29,30,31]. This may partly account for the trans-coelomic pattern of dissemination of endometriosis. Here, we showed that cell migration occurs in response to molecular cues emitted from O-ASC, and HGF/c-MET signaling is a pathway mediating this.

HGF/c-MET signaling is not well-studied in endometriosis. Yoshida et al. [32] described high levels of HGF in the peritoneal fluid of endometriosis patients. We have shown here that the omentum is a source of HGF. The level of HGF produced by the omentum of endometriosis patients was similar to that in CCC, EC and MC subtypes of ovarian cancer. Moreover, endometriosis cells expressed c-MET at a level similar to ovarian cancer cells, suggesting that endometriosis cells have a similar capacity to migrate and disseminate in the abdominopelvic cavity. Importantly, we demonstrated that O-ASC CM-induced cell migration could be curtailed in endometriosis with c-MET inhibition. Therefore, therapies that disrupt HGF/c-MET signaling could be considered for further investigation in endometriosis patients, with the aim of retarding or halting disease progression. In fact, an omentectomy during endometriosis surgery is worthy of consideration for clinical trial, testing the hypothesis that removal of the omentum would diminish its pro-migratory stimulus, leading to better disease control, thus obviating the need for hormonal therapy (current non-surgical treatment for endometriosis).

Several therapeutics, including small molecules and antibodies, such as foretinib, cabozantinib and YYB-101, have been developed that target the HGF/c-MET pathway and have been tested in clinical trials in ovarian cancer patients, yet none has shown significant efficacy [22]. Patient selection may be the issue for the lack of efficacy. Perhaps future clinical trials investigating c-MET inhibitors and anti-HGF therapies in ovarian cancer should select for HGSC patients, given that the omenta in these patients produced significantly greater amounts of HGF compared to other subtypes and induced a robust cell migration response in HGSC.

The omental secretome profiles are heterogeneous. First, the omenta from endometriosis patients produced higher IL-8 levels, compared to ovarian cancer. IL-8 is a pro-inflammatory cytokine, and endometriosis is associated with inflammation and fibrosis [33]. Arici et al. observed elevated IL-8 levels in the peritoneal fluid of endometriosis patients that correlated with disease severity [34]. While the cause-and-effect relationship is unknown, we surmise that IL-8 from the omenta of endometriosis patients creates a conducive inflammatory environment for disease progression, thereby a potential candidate for therapeutic development and intervention. Second, the omenta of HGSC patients produced higher levels of three cytokines, namely HGF, SDF-1a/CXCL12 and MCP-1/CCL2, known to mediate cell migration, invasion, proliferation and metastasis in ovarian cancer [21,23,24,25,35,36,37]. Measuring these cytokine levels in the peritoneal fluid (ascites, peritoneal lavage) of HGSC patients potentially serves as a method of selecting patients for treatments targeting HGF, SDF-1a and MCP-1 pathways. Third, the omenta from patients with endometriosis and benign cyst produced lower levels of VEGF-A and IL-6 compared to ovarian cancer. VEGF-A is an angiogenic factor [38], and IL-6 is a pro-inflammatory cytokine that promotes cell proliferation [39].

Although the omenta of HGSC patients did not produce the highest level of VEGF, the O-ASC CM from HGSC patients consistently gave rise to high levels of angiogenesis. This suggests the presence of other pro-angiogenic factors in O-ASC CM from HGSC. O-ASC CM from EMS and BorMT patients induced a fair amount of angiogenic activity while the O-ASC CM from the patient with BeCy did not exhibit angiogenic potential. Of note, due to the small sample size, BeCy and MC had a single data point and BorMT had only two data points, the results from these histological types may not be representative and should be interpreted with caution. Whether the angiogenic effects of the omentum’s secreted factors is a marker of malignant potential remains undetermined.

Given the omental secretome profiles and the presence of factors known to induce cell proliferation, e.g., HGF, MCP-1 and FGF-2, an increase in cell proliferation upon exposure to O-ASC CM was expected. However, we observed no significant change in cell proliferation in the primary cell cultures of endometriosis and ovarian cancer. This suggests the presence of proliferation inhibitors in O-ASC CM, or the cells lack the cognate receptors to respond to proliferative stimuli. Another explanation is that cell migration, angiogenesis and cell proliferation are governed by distinct and separate cell-intrinsic programs, and the human omentum emits factors that functionally induce migration and angiogenesis (in different circumstances) but not proliferation.

PTTG1 (securin) was significantly upregulated in the migrated population of HGSC, EC and EMS, i.e., across three different histological types, in both malignant and benign conditions. PTTG1 is relatively unexplored in endometriosis, although it is known to be overexpressed in several tumors and implicated in various tumor processes [40,41,42,43,44,45,46,47]. In ovarian cancer, inhibition of PTTG1 suppressed cell proliferation and tumor growth in nude mice model [45,48]. Given that PTTG1 is expressed in the migration population of EMS, similar to ovarian cancer, inhibition of PTTG1 may delay the progression of endometriosis and ovarian cancer. This is a molecule worthy of further study for prognostic and therapeutic purposes.

In our study, we mainly used the transwell migration assay to examine the cell migration ability of endometriosis and ovarian cancer in response to O-ASC CM. Although the assay is simple and easy to setup, there are also several limitations. Firstly, the setup is two-dimensional and non-physiological, which does not accurately reflect the endogenous 3D microenvironment in the abdominopelvic cavity. Secondly, the assay may miss out other cell–cell interactions present in the 3D microenvironment that could potentially affect the migration dynamics. Lastly, the transwell migration assay is an endpoint assay, which does not capture cell migration in real time. Another limitation of this study is the short lifespan of primary cell cultures, which restricted the type of experiments to those with a short time course. Often, we could not repeat the experiment with the same primary culture. We circumvented this by repeating the experiment in multiple primary cultures of the same histology, assuming that they have similar biological backgrounds. Additionally, challenges in tissue acquisition resulted in a small sample size for certain tissues, e.g., omentum from endometriosis patients. This is largely because very few endometriosis patients consented to omental sampling and the resection of the omentum is not a standard procedure for endometriosis surgery. Moreover, endometriosis surgery is usually performed laparoscopically, which makes sampling of the omentum technically challenging, further reducing the chances of obtaining omental tissue.

## 4. Materials and Methods

### 4.1. Clinical Specimens

Approvals for this study were obtained from the SingHealth Centralised Institutional Review Board, encompassing the National Cancer Centre Singapore, Singapore General Hospital and Kandang Kerbau Hospital for Women and Children (CIRB 2015/2595). Inclusion criteria were patients who had symptoms, e.g., abdominal bloatedness or pain, and signs, e.g., image evidence of intra-abdominal masses that were suspicious for malignancy or endometriosis. Exclusion criteria were patients who were not medically fit to undergo surgery. Tissue samples were obtained from patients with informed consent. As per local institutional clinical routine, patients with ovarian cancer underwent open surgery, whereas patients with endometriosis underwent laparoscopic or open surgery. Tissue samples of omentum, endometriosis and ovarian cancer were obtained ex vivo, dissected fresh by the pathologist. The tissues were collected in RPMI1640 medium and delivered on ice to the laboratory within one hour of collection.

### 4.2. Primary Cell Culture of Human Ovarian Cancer and Endometriosis

Tissue samples of ovarian cancer and endometriosis were cut into ~1–2 mm^3^ pieces using sterile scalpels. Single-cell suspensions were obtained using the Tumor Dissociation Kit (Miltenyi Biotec, Bergisch Gladbach, Germany, #130-095-929), filtered through a 70 μm cell strainer and centrifuged for 5 min at 600× *g*. Red blood cells were removed using 5 mL of RBC Lysis Buffer (G-Biosciences, St. Louis, MO, USA, #786-672) and incubated at room temperature for 10 min. The cell pellets were washed with phosphate-buffered saline (PBS) and re-suspended in MCDB 105/M199 (1:1) medium, supplemented with 10% fetal bovine serum (FBS) and 1% penicillin–streptomycin. Cultures were maintained in a humidified incubator at 37 °C with 5% CO_2_.

### 4.3. Primary Cell Culture of Human Omentum

Tissue samples of human omentum were cut into ~1–2 mm^3^ pieces using sterile forceps and scissors. They were digested with Type I collagenase (1 mg/mL; Stemcell Technologies, Vancouver, BC, Canada, #07416) in Dulbecco’s Modified Eagle/Nutrient Mixture F-12 (DMEM/F12) media for 1 h with shaking at 37 °C. Collagenase was inactivated by the addition of equal volume of Dulbecco’s Modified Eagle Medium F-12 (DMEM/F12) with 10% FBS. Undigested tissue was removed by filtering through a nylon mesh strainer (250 μm, UV-irradiated, Pierce Tissue Strainer; Thermo Fisher Scientific, Waltham, MA, USA, #87791). The filtrate was centrifuged for 5 min at 200× *g*. Red blood cells were removed using 5 mL of RBC Lysis Buffer (G-Biosciences #786-672) and incubated at room temperature for 10 min. The supernatant, containing mature adipocytes, was removed. Cell pellets, containing omental-adipose stromal cells (O-ASCs), were washed in PBS before being re-suspended in DMEM/F12 medium, supplemented with 10% FBS, 1% antibiotic–antimycotic, 1% MEM non-essential amino acids and incubated at 37 °C with 5% CO_2_.

### 4.4. Adipogenic Differentiation of O-ASC

To confirm the adipogenic potential of O-ASCs, they were grown in induction media, comprising DMEM/F12 media, supplemented with 2% FBS, 15 mM HEPES, 15 mM NaHCO_3_, 33 μM biotin, 5.5 μg/mL transferrin, 10 μg/mL insulin, 1 μM dexamethasone (Sigma-Aldrich, St. Louis, MO, USA, #D4902), 0.5 mM IBMX (3-isobutyl-1-methyl-xanthine) (Sigma-Aldrich, #I7018) and 1% antibiotic–antimycotic for 10 to 14 days. The differentiated cells were then examined for the presence of lipid droplets with Oil Red O staining. They were fixed in 10% formalin, followed by the addition of 2 mL of freshly prepared Oil Red O working solution (Sigma-Aldrich, #MAK194) and kept at room temperature for 20 min. Deionized water was used to remove excess stain before microscopic examination.

### 4.5. Generation of O-ASC Conditioned Medium

After induction of adipogenesis, the differentiated O-ASCs were kept in DMEM/F12 medium, supplemented with 5% FBS (5% FBS-DMEM/F12) for 48 h at 37 °C in a 5% CO_2_ incubator. The medium collected after 48 h was used as conditioned medium (CM), filtered through 0.22 µm filters (Millipore, Burlington, MA, USA) and stored at −80 °C for further use.

### 4.6. Transwell Cell Migration Assay

Migration assays were performed using a 2-compartment Boyden chamber with a physical barrier of an 8 μm polycarbonate membrane transwell insert (Corning, NY, USA, #3422). About 1.0–2.5 × 10^4^ serum-free starved cells that were primary cultures of either ovarian cancer or endometriosis were introduced to the upper chamber. The lower chamber contained either CM or control media (5% FBS-DMEM/F-12).

After incubating for 24 h at 37 °C, cells that had migrated were fixed and stained with 0.5% crystal violet in 25% methanol for 10 min and quantified by counting the number of cells in five random microscopic fields (magnification ×40 or ×100) per membrane. The migration assays were performed in duplicate. The ImageJ program (https://imagej.nih.gov/ij/, accessed on 28 March 2023) was used to quantify the number of migrated cells.

In c-MET inhibition migration experiments, cells were pre-treated with or without the c-MET inhibitor (2 μM PHA665752; Sigma-Aldrich, #PZ0147) for 1 h and then washed with PBS. In each migration assay, 1.0–2.5 × 10^4^ of treated or untreated cells were plated in the upper chamber; the medium in the lower chamber was either CM or control media.

### 4.7. Wound Healing Assay

Ovarian cancer or endometriosis cells were cultured to 100% confluence in each chamber of ibidi culture-insert placed in a 35 mm μ-Dish (ibidi GmbH, Gräfelfing, Germany). The culture insert was removed from the dish using a pair of sterilized tweezers to generate a 500 μm wound. Cells were gently rinsed twice with PBS, before O-ASC CM or 5% FBS-DMEM/F12 control media were added. Cells were incubated for 2 days to allow them to migrate into the wound. Five random fields (40× magnification) of the wound images were captured under a microscope at the starting time point of 0 h, 24 h and 48 h to compare the size of healed areas.

### 4.8. Luminex Assays

O-ASC CM was collected and underwent protein profiling using Luminex xMAP Technology, with the Immune Monitoring 65-Plex Human ProcartaPlex™ Panel (Invitrogen, Waltham, MA, USA, #EPX650-10065-901). Aliquots of 80 μL were placed in a 96-well plate for MAP antigen analysis, according to the manufacturer’s protocols. Assays were run using the Flexmap 3D system (Luminex, Austin, TX, USA) and analyzed with Bio-plex Manager software (version 6.2) (Bio-Rad, Hercules, CA, USA). Data outside the range of the standards, but within the asymptote of the equation, were extrapolated beyond the standard curve. For negative control, 5% FBS-DMEM/F-12 medium was used.

### 4.9. Transcriptome Profiling

Total RNA was isolated using RNeasy Mini Kit (Qiagen, Hilden, Germany), according to manufacturer’s protocol. After assessing RNA quality, 50 ng total RNA from each sample was analyzed on the nCounter^®^ SPRINT platform using the PanCancer Progression Panel (NanoString, Seattle, WA, USA). RNA expression data were normalized using nSolver Advance Analysis (v2.0.134) and ROSALIND (https://www.rosalind.bio/, accessed on 16 November 2023) softwares. Differentially expressed genes were identified using a cut-off of absolute fold change ≥ 1.6 and *p* < 0.05. Heatmaps and volcano plots were generated using pheatmap (v1.0.12) and tidyverse (v2.0.0) packages in R (v4.0.2), respectively.

### 4.10. Quantitative Real-Time Polymerase Chain Reaction (qRT-PCR) Analysis

Total RNA was extracted from cells using RNeasy Mini Kit (Qiagen). Reverse transcription was performed on 1 μg of total RNA, using the SuperScriptIV^TM^ First Strand Synthesis Kit, according to manufacturer’s protocol (Invitrogen, #18091050). RT-qPCR was performed using ABI 7500 Fast Real Time PCR System (Applied Biosystems, Waltham, MA, USA) with PowerUP SYBR Green Master Mix (Applied Biosystems). Reactions were performed in duplicate. Human *GAPDH* was used to normalize gene expression levels. Fold change was calculated relative to a benign ovarian cyst, acting as control. The sequences of primers were as follows: *c-MET*, forward: ATGAGCACTGCTTTAATAGGACAC, reverse: GGACTTCGCTGAATTGACCC; *GAPDH*, forward: CAAGCTCATTTCCTGGTATGAC, reverse: CAGTGAGGGTCTCTCTCTTCCT.

### 4.11. Angiogenesis Assay

Human umbilical vein endothelial cells (HUVECs, pooled donors; Lonza, Basel, Switzerland, #C2519AS) were maintained in Endothelial Basal Medium-2 (EBM-2) with supplements (Lonza, #CC-3162), according to manufacturer’s protocol. To assess the angiogenic potential of O-ASC CM, HUVECs (1.1 × 10^4^) were added to 96-well plates pre-coated with Matrigel (8.4 mg/mL; Corning #354230) and exposed to O-ASC CM. After 16 h of incubation at 37 °C in the presence of O-ASC CM or control medium (5% FBS-DMEM/F12), the tube formation ability of HUVECs was measured; five replicates of each experiment was performed. The tube-like network was visualized with the Nikon Eclipse TS300 microscope and photographed. The extent of angiogenesis was analyzed based on different parameters, such as the number of junctions, number of meshes and total tubule length, using ImageJ software (https://imagej.nih.gov/ij/, accessed on 5 October 2023).

### 4.12. Cell Proliferation Assays

Cell proliferation was evaluated using the MTT cell viability assay (CyQUANT, Thermo Fisher Scientific, #V13154), according to the manufacturer’s instructions. For experiments interrogating the cell proliferative effects of O-ASC CM, cells from primary cultures of either ovarian cancer or endometriosis were plated in 96-well plates (1.5–3.2 × 10^3^ cells/well) in growth media containing 10% FBS and incubated overnight at 37 °C. The media was then replaced with either 5% FBS-DMEM/F12 control media or O-ASC CM diluted with 5% FBS-DMEM/F12 in a 1:1 ratio. The media was changed after 3 days and 6 days of incubation at 37 °C. MTT assay measurements were taken in duplicate for up to 8 days.

### 4.13. Statistical Analysis

In vitro data were analyzed with GraphPad Prism v9.1.0 software. Samples were analyzed in replicate for migration, proliferation, RT-qPCR and angiogenesis assays. Statistical significance was assessed by unpaired student’s *t*-test, Mann Whitney U test or ANOVA followed by Tukey’s multiple comparison test, where results were expressed as mean ± standard deviation (SD), and *p*-values < 0.05 were considered significant.

## 5. Conclusions

In summary, human omental-adipose stromal cells (O-ASC) secrete a panoply of factors that induce cell migration and angiogenesis in endometriosis and ovarian cancer. HGF and VEGF-A are present in the O-ASC CM of ovarian cancer and endometriosis, albeit at variable levels. PTTG1 is expressed in migrated cell populations in both endometriosis and ovarian cancer. The omentum provides a favorable environment for trans-coelomic spread of endometriosis and ovarian cancer. These findings have provided leads for future work in improving clinical outcomes of endometriosis and ovarian cancer patients.

## Figures and Tables

**Figure 1 ijms-26-01822-f001:**
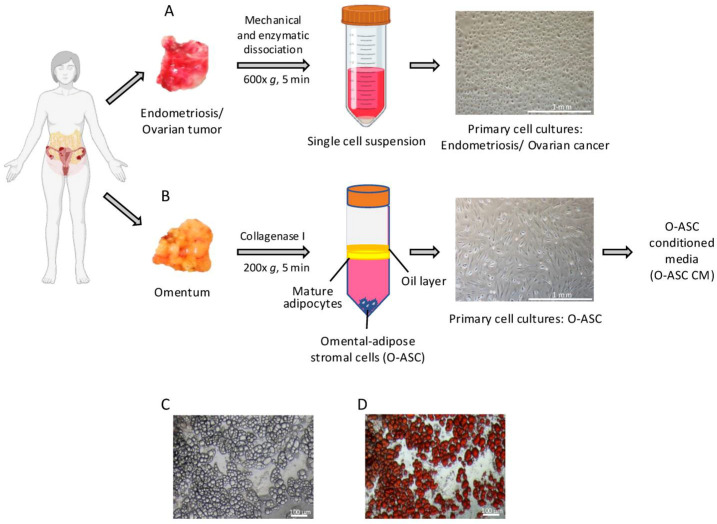
Developing primary cell cultures of endometriosis, ovarian tumors and the omentum from fresh human tissues, and obtaining conditioned media from omental-adipose stromal cells (O-ASC). (**A**) Processing of fresh human ovarian tumors and endometriosis. The tissues were mechanically and enzymatically dissociated to form single cell suspensions before expansion in culture. An example of ovarian cancer cell culture from patient E335 at passage 3 was shown, 40× magnification. (**B**) Processing of fresh human omental tissue. The tissues were enzymatically digested and separated via centrifugation. An example of O-ASC culture from patient E255 at passage 3 was shown, 40× magnification. Conditioned media (CM) was collected from O-ASC cultures post-adipogenic differentiation. (**C**,**D**) O-ASC after adipogenic differentiation, before (**C**) and after (**D**) Oil-Red O staining of lipid droplets, 100× magnification. The figure was partially created with BioRender.com.

**Figure 2 ijms-26-01822-f002:**
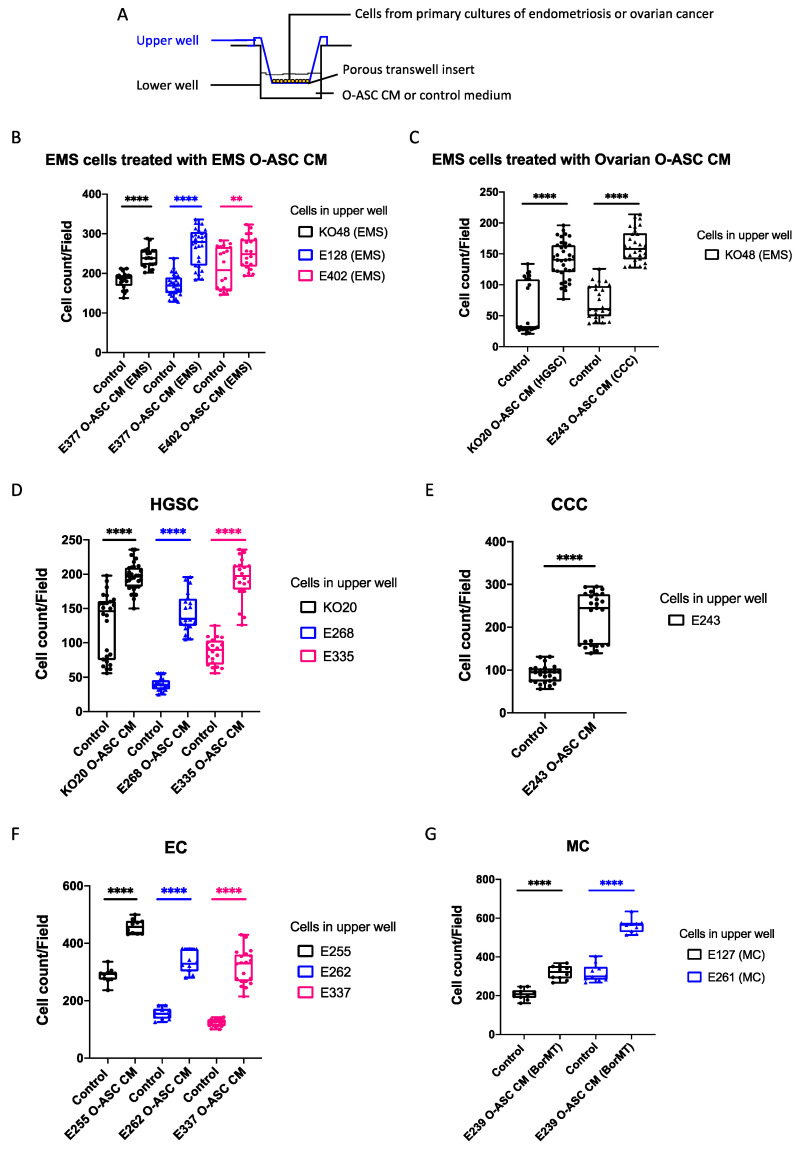
Omental-adipose stromal cell conditioned media (O-ASC CM) increased cell migratory activity in primary cultures of endometriosis and ovarian cancer using the transwell migration assay. (**A**) Transwell cell migration set-up using the 2-compartment Boyden chamber. (**B**–**G**) For each panel, the x-axis shows the type of medium used, the y-axis denotes the number of migrated cells, the legend indicates cells used in the upper well. (**B**) O-ASC CM derived from endometriosis (EMS) cases increased the migration activity of endometriosis cells. KO48 and E128 cells were assayed with O-ASC CM from E377. E402 cells was assayed with O-ASC CM from E402. (**C**) O-ASC CM derived from ovarian cancer cases, HGSC (KO20) and CCC (E243), increased the migration activity of endometriosis cells (KO48) more robustly than O-ASC CM derived from endometriosis patients (see (**B**)). (**D**–**G**) O-ASC CM derived from ovarian cancer histological subtypes increased the migration activity of ovarian cancer cells, as shown in (**D**): HGSC (KO20, E268, E335), (**E**): CCC (E243), (**F**): EC (E255, E262, E337) and (**G**): MC (E127, E261). The O-ASC CM used in each assay was derived from the same patient except in both MC cases, where the O-ASC CM was derived from a BorMT patient (E239). Unpaired *t*-test was used for statistical comparison (** *p* < 0.01; **** *p* < 0.0001). See also Appendix A. For each experiment, five random fields were imaged per transwell and at least two replicates were included. Abbreviations: HGSC, high grade serous carcinoma; CCC, clear cell carcinoma; EC, endometrioid carcinoma; MC, mucinous carcinoma; BorMT, borderline mucinous tumor; EMS, endometriosis.

**Figure 3 ijms-26-01822-f003:**
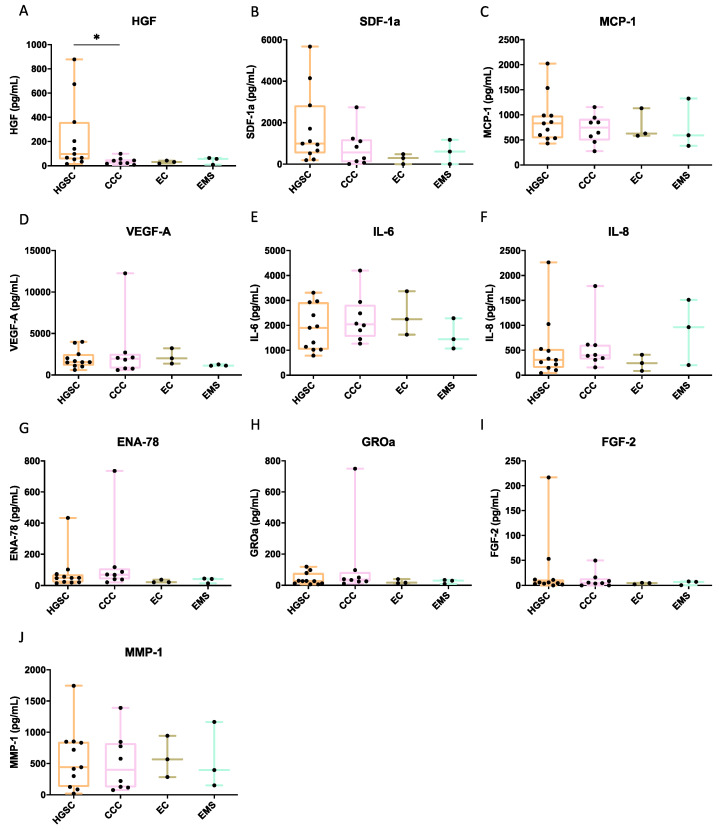
Cytokine profiles of omental-adipose stromal cell conditioned media (O-ASC CM) derived from patients with malignant ovarian tumors and benign endometriosis using Luminex assay. Quantification of (**A**) HGF, (**B**) SDF-1a, (**C**) MCP-1, (**D**) VEGF-A, (**E**) IL-6, (**F**) IL-8, (**G**) ENA-78, (**H**) GROa, (**I**) FGF-2 (**J**) MMP-1 in O-ASC CM derived from patients with high grade serous carcinoma (HGSC), *n* = 11; clear cell carcinoma (CCC), *n* = 8; endometrioid carcinoma (EC), *n* = 3; endometriosis (EMS), *n* = 3. Mann-Whitney U test was used for statistical comparison between the different histological types. (* *p* < 0.05). Due to small sample size, the cytokine profiles of O-ASC CM derived from patients with MC, BorMT and BeCy are not shown here, see also Appendix A and Appendix A. Abbreviations: HGSC, high grade serous carcinoma; CCC, clear cell carcinoma; EC, endometrioid carcinoma; MC, mucinous carcinoma; BorMT, borderline mucinous tumor; EMS, endometriosis.

**Figure 4 ijms-26-01822-f004:**
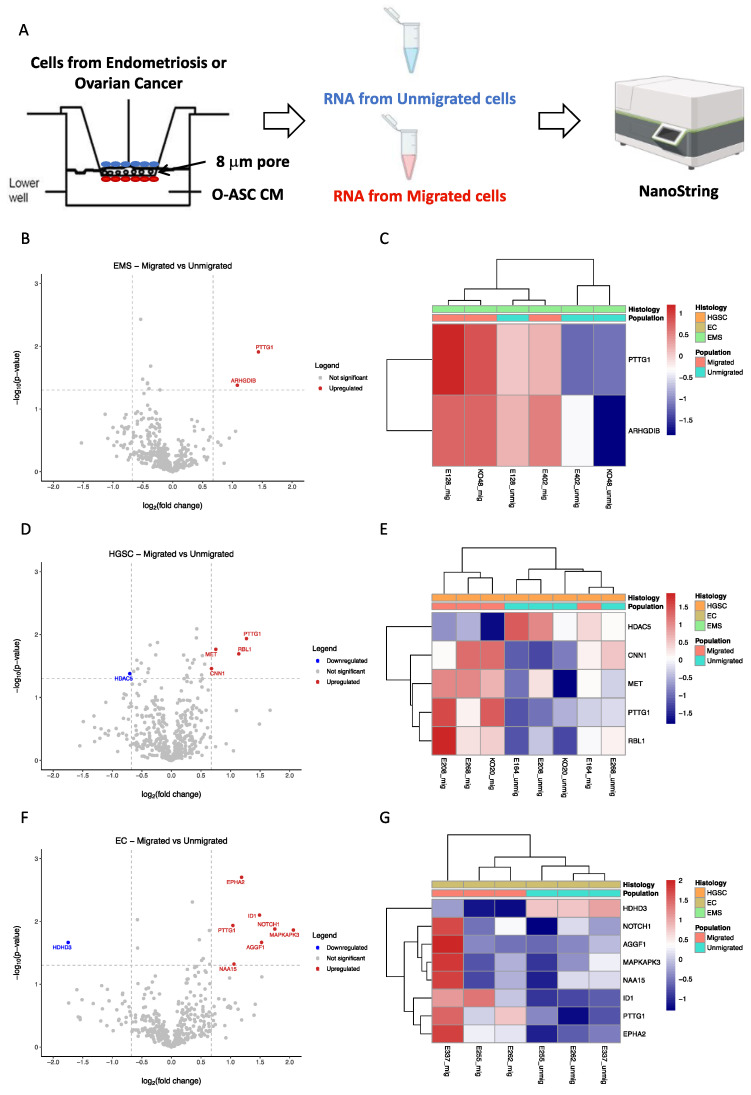
Differentially expressed genes between migrated and unmigrated cell populations of endometriosis and ovarian cancer in the presence of omental-adipose stromal cell conditioned media (O-ASC CM). (**A**) Schematic diagram of the experimental set-up. The O-ASC CM used in the lower well were histologically-matched with cells added in the upper well (KO20 O-ASC CM with HGSC cells, *n* = 4; E337 O-ASC CM with EC cells, *n* = 3; E377 O-ASC CM with EMS cells, *n* = 3) (**B**–**G**) Volcano plots and heatmaps showing the differentially expressed genes between migrated and unmigrated cell populations of EMS (**B**,**C**), HGSC (**D**,**E**) and EC (**F**,**G**). On the volcano plots, differentially expressed genes with *p* < 0.05 and fold change ≥ 1.6 are depicted in red (up-regulated) and blue (down-regulated). On the heatmaps, red represents highly-expressed genes while blue represents lowly-expressed genes. The figure was partially created with BioRender.com. See also Appendix A. Abbreviations: HGSC, high grade serous carcinoma; EC, endometrioid carcinoma; EMS, endometriosis.

**Figure 5 ijms-26-01822-f005:**
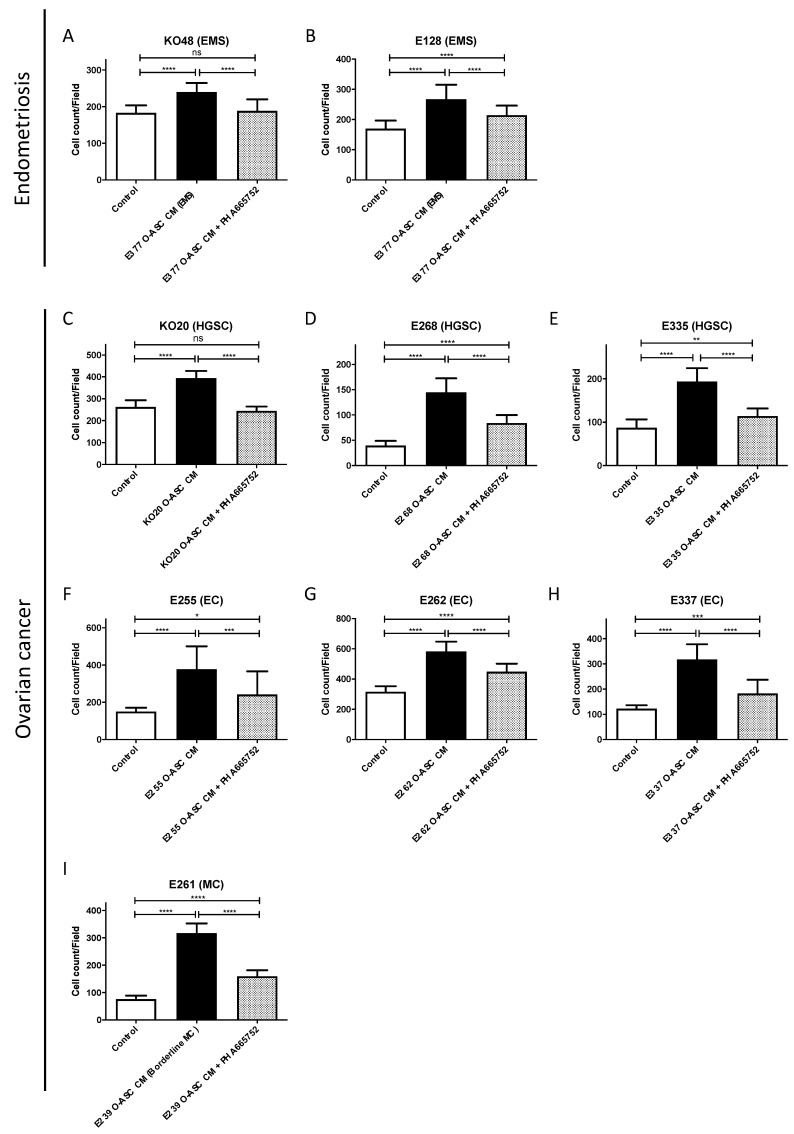
c-MET inhibition abrogated omental-adipose stromal cell conditioned media (O-ASC CM)-induced cell migration in endometriosis and ovarian cancer. For each assay, the chart title states the type of cells studied, the x-axis shows the type of medium used, the y-axis denotes the number of migrated cells. O-ASC CM used for each assay was derived from the same patient, except in the MC and EMS cases, where the O-ASC CM was derived from a BorMT patient (E239) and EMS patient (E377), respectively. Cell migration in ovarian cancer and endometriosis was enhanced in the presence of O-ASC CM but inhibited in the presence of c-MET inhibitor, PHA665752. This was observed across all histological types, including EMS (**A**,**B**), HGSC (**C**–**E**), EC (**F**–**H**) and MC (**I**). ANOVA followed by Tukey’s multiple comparison test was used for statistical comparison. Results are shown as mean + standard deviation (SD) (NS: Not significant; * *p* < 0.05; ** *p* < 0.01; *** *p* < 0.001; **** *p* < 0.0001). See also Appendix A. For each experiment, five random fields were imaged per transwell and at least two replicates were included. Abbreviations: HGSC, high grade serous carcinoma; EC, endometrioid carcinoma; MC, mucinous carcinoma; BorMT, borderline mucinous tumor; EMS, endometriosis.

**Figure 6 ijms-26-01822-f006:**
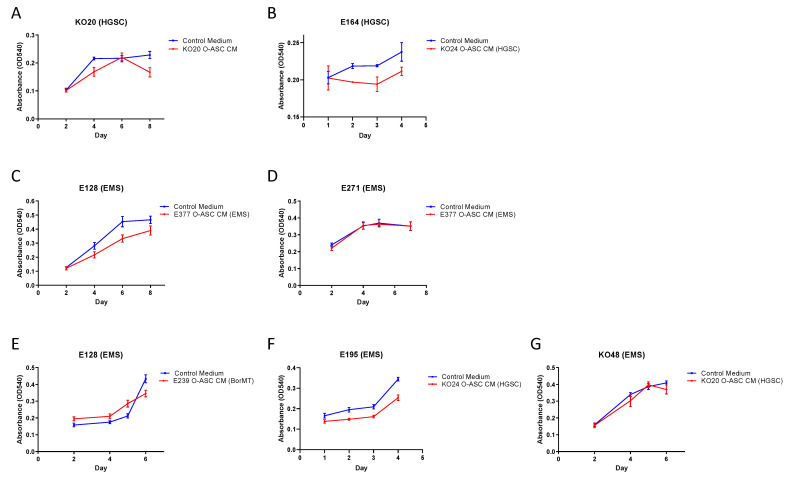
Effect of omental-adipose stromal cell conditioned media (O-ASC CM) on cell proliferation in endometriosis and ovarian cancer. For each assay, the chart title states the type of cells studied, the chart legend shows the O-ASC CM used. (**A**,**B**) O-ASC CM from HGSC patients (KO20, KO24) had no significant effect on cell proliferation in HGSC (KO20, E164). (**C**,**D**) O-ASC CM from EMS patient (E377) had no significant effect on cell proliferation in EMS (E128, E271). (**E**–**G**) O-ASC CM from BorMT and HGSC patients (E239, KO20, KO24) had no significant effect on cell proliferation in EMS (E128, E195, KO48). For each experiment, at least two replicates were included. Error bars represent standard deviation (SD). Abbreviations: HGSC, high grade serous carcinoma; BorMT, borderline mucinous tumor; EMS, endometriosis.

## Data Availability

Data from this study are available upon reasonable request to the corresponding author.

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
