# Peer review of "Cell Migration in Endometriosis Responds to Omentum-Derived Molecular Cues Similar to Ovarian Cancer"

_ijms, 2025, doi:10.3390/ijms26051822_

Round 1

Reviewer 1 Report

Comments and Suggestions for Authors

The authors have done an excellent job of presenting their findings clearly and effectively. The overall structure and content are strong, with only minor corrections to improve clarity and consistency. Once these minor adjustments are made, the work will be even more polished and ready for publication.

- Lines 220 until 228. Please correct the gene names. These must be written in italics.

- Line 318. There is a number (1) in proliferation. If this is a reference, please correct it. 

More Detailed Comments

The well-conceived study contributes meaningfully to understanding the shared migratory pathways in endometriosis and ovarian cancer. Addressing the noted gaps will significantly enhance its impact and clarity.

*Title and Abstract: The title accurately reflects the study's focus on the relationship between omentum-derived molecular cues and cell migration in endometriosis and ovarian cancer. The abstract effectively summarizes the study's objectives, methods, findings, and significance.

*Introduction: The introduction provides a solid background on endometriosis, ovarian cancer, and the biological role of the omentum. It establishes the connection between these diseases and trans-coelomic migration.

*Methods:
- The study includes a comprehensive description of the experimental workflows, which encompass primary cell cultures, migration assays, and transcriptome profiling.
- The adherence to ethical guidelines and the inclusion of patient consent details are commendable.

*Results:
- Results are presented with adequate figures, including migration assays and cytokine profiles.
- The upregulation of PTTG1 across conditions is a novel finding supported by robust gene expression data.

*Discussion:
- The discussion links the findings to clinical implications, including potential targeted therapies such as c-MET inhibitors.
- The relevance of PTTG1 as a biomarker for both diseases is well-articulated.
- Sometimes, the discussion becomes speculative, especially concerning the therapeutic potential of omentectomy in endometriosis, which lacks substantial experimental support.
- Are there any available PTTG1 and/or c-MET inhibitors that could treat endometriosis and cancer?

*Figures and Tables: Figures and tables are well-annotated.

*References: The references are current and relevant.

*Overall Significance and Final recommendations:
- The study provides significant insights into the molecular mechanisms underlying cell migration in endometriosis and ovarian cancer.
- It identifies potential therapeutic targets (HGF/c-MET and PTTG1).
- Discuss the clinical implications of findings and the therapeutic potential of targeting the HGF/c-MET and PTTG1 pathways.

Author Response

Response to Comments from Reviewer 1

Thank you very much for taking the time to review this manuscript. Please find the detailed responses in blue below and the corresponding revisions are highlighted/in track changes in the resubmitted files.

Comment 1: Lines 220 until 228. Please correct the gene names. These must be written in italics.
Response 1: Thank you for pointing this out. We have italicized the gene names in lines 269 to 282.

Comment 2: Line 318. There is a number (1) in proliferation. If this is a reference, please correct it.
Response 2: Thank you for pointing this out. We have removed (1) and added in the correct reference (ref 39) at line 423.

Comment 3: Sometimes, the discussion becomes speculative, especially concerning the therapeutic potential of omentectomy in endometriosis, which lacks substantial experimental support.
Response 3: Thank you for the comment. We agree with the reviewer that the therapeutic potential of an omentectomy in endometriosis is speculative, and have qualified this hypothesis-generating statement with 'worthy of consideration for clinical trial' (line 378) and 'testing the hypothesis' (line 379). 

Comment 4: Are there any available PTTG1 and/or c-MET inhibitors that could treat endometriosis and cancer?
Response 4: Thank you for the comment. There are available c-MET inhibitors but no available PTTG1 inhibitors. We have added the following paragraph in Discussion (lines 382-389):

‘Several therapeutics, including small molecules and antibodies, such as foretinib, cabozantinib and YYB-101, have been developed that target the HGF/c-MET pathway, and have been tested in clinical trials in ovarian cancer patients, yet none has shown significant efficacy [22]. Patient selection may be the issue for the lack of efficacy. Perhaps future clinical trials investigating c-MET inhibitors and anti-HGF therapies in ovarian cancer should select for HGSC patients, given that the omenta in these patients produced significantly greater amounts of HGF compared to other subtypes, and induced a robust cell migration response in HGSC.”

Comment 5: Discuss the clinical implications of findings and the therapeutic potential of targeting the HGF/c-MET and PTTG1 pathways.
Response 5: In the Discussion section, the clinical implications and therapeutic potential of targeting the HGF/c-MET and PTTG1 are discussed as follows:

Targeting HGF/c-MET in endometriosis: 'Therefore, therapies that disrupt HGF/c-MET signaling could be considered for further investigation in endometriosis patients, with the aim of retarding or halting disease progression. In fact, an omentectomy during endometriosis surgery is worthy of consideration for clinical trial, testing the hypothesis that removal of the omentum would diminish its pro-migratory stimulus, leading to better disease control, thus obviating the need for hormonal therapy (current non-surgical treatment for endometriosis). ' (lines 376-381)

Targeting HGF/c-MET in ovarian cancer: ‘Several therapeutics, including small molecules and antibodies, such as foretinib, cabozantinib and YYB-101, have been developed that target the HGF/c-MET pathway, and have been tested in clinical trials in ovarian cancer patients, yet none has shown significant efficacy [22]. Patient selection may be the issue for the lack of efficacy. Perhaps future clinical trials investigating c-MET inhibitors and anti-HGF therapies in ovarian cancer should select for HGSC patients, given that the omenta in these patients produced significantly greater amounts of HGF compared to other subtypes, and induced a robust cell migration response in HGSC.” (lines 382-389)

Targeting PTTG1 in ovarian cancer and endometriosis: ‘In ovarian cancer, inhibition of PTTG1 suppressed cell proliferation and tumor growth in nude mice model [45,48]. Given that PTTG1 is expressed in the migration population of EMS, similar to ovarian cancer, inhibition of PTTG1 may delay the progression of endometriosis and ovarian cancer. This is a molecule worthy of further study for prognostic and therapeutic purposes.’ (lines 444-449)

Reviewer 2 Report

Comments and Suggestions for Authors

This study explored the role of secreting factors from the human omentum in intra-abdominal cell  migration of endometriosis and ovarian cancer cells. Although the paper is of great interest, there are major issues with statistical approach, that authors should consider correcting prior to acceptance.

Please see comments on statistics below, and also some minor details:

Line 45 in the Abstract: instead of “progression of these conditions” put “abdomen-pelvic/peritoneal spread of endometriosis and ovarian cancer.”

Line 77 in the Introduction. Add more information about omentum function such as immune regulation that enable peritoneal defence mechanisms, forming dense adhesions at the sites of contamination or wounding, pro- angiogenic activity etc. Also add that due to its intrinsic angiogenic properties it is a frequent site of metastatic disease for many malignancies (PMID: 11819552).

Figure 2 should be representative of the entire groups of patients not only  selected results for endometriosis and  for ovarian cancer. It should be shown as a box plot to depict the distribution of all individual values for control and a box plot for all the O-ASC CM-exposed. One panel should represent the endometriosis cells treated with autologous  O-ASC CM, and another box plot should represent endometriosis cells treated with ovarian cancer O-ASC CM. Further, in the second panel, it should present separate box plots with cases of HGSC,CCC, EC and MC cells treated with respective O-ASC CM.

Line 172 do you mean matrix metalloproteinase instead of metallopeptidase ?

Characterizing the secretome of omental-adipose stromal cells from the 29 samples of O-ASC CM is problematic due to numbers of samples of subtypes: malignant (11 HGSC, 8 CCC, 3 EC, 1 MC), to borderline tumors (2 BorMT), endometriosis (3 EMS) and benign cyst (1 BeCy). The types where you have at least 3 individual samples could be retained in the analysis, but having only 1 for MC and BeCy and 2 for BorMT is not enough for these analyses. I would recommend excluding them from these analyses. Also, cytokines were consistently more highly secreted in HGSC. Since you have a much greater sample size for HGSC compared to other subtypes, this could influence the analysis.

In line 202 you state that cells were each exposed  to histologically matched O-ASC CM but that is not correct since in the bracket you state that a single sample E377 was used for the treatment of all EMS and EC cells. Also, a single KO20 O-ASC CM was used for the treatment of all HGSC cells from 4 donors. Thus, you have only 2 different conditioned media for the treatment of all cell samples.

To determine the differences in HGF/c-MET signaling in Control, O-ASC CM treated group and inhibitor+ O-ASC CM group, ANOVA should be used instead of t test.

Line 266 instead of cell proliferation MTT rather analyses cell viability, as it represents the number of metabolically active cells. Include the MTT results from supplementary in the main manuscript as you describe these results in the section 266-271. Also, add explanation why you evaluated cell viability for 6 days.

Line 278 “primary cultures of endometriosis, ovarian  cancer” correct to “primary cultures established from the tissue samples of endometriosis and ovarian cancer patients”

Line 278-283 should be rephrased, it is not clear what the authors meant in these parts: “maximally reflect their usual functional states,” and “tissue’s microcosm of cell subpopulations in vivo, which may positively influence their  longevity ex vivo”

Line 317 there is a reference no 1 in superscript at the end of the sentence.

Discussion should be improved by citing some other findings relevant to be compared with your results.

Lines 318-321 are generalized statements that do not provide any clear conclusion.

Line 324” Notably, the O-ASC CM from the patient with BeCy did not exhibit angiogenic potential.” Since you had only a single sample of BeCy O-ASC CM it is very likely that by increasing sample size, some differences could be observed. Add here the limitations of this study by referring to number of samples.

Line 342 “inhibition of PTTG1 may delay the progression or recurrence of endometriosis and ovarian cancer” is there a reference for this statement or this is an author’s assumption? If it is an assumption, add explanations, since you did not evaluate the effects of PPTG1 expression inhibition.

Improve the explanation of the limitations of this study in the final paragraph, as they are poorly written.

Author Response

Response to Comments from Reviewer 2

Thank you very much for taking the time to review this manuscript. Please find the detailed responses in blue below and the corresponding revisions are highlighted/in track changes in the resubmitted files.

Comment 1: Line 45 in the Abstract: instead of “progression of these conditions” put “abdomen-pelvic/peritoneal spread of endometriosis and ovarian cancer.”
Response 1: Thank you for the suggestion. We have updated the abstract according to your suggestion in lines 45-46.

Comment 2: Line 77 in the Introduction. Add more information about omentum function such as immune regulation that enable peritoneal defence mechanisms, forming dense adhesions at the sites of contamination or wounding, pro- angiogenic activity etc. Also add that due to its intrinsic angiogenic properties it is a frequent site of metastatic disease for many malignancies (PMID: 11819552).
Response 2: Thank you for the suggestion. We have included more information on the omentum function in the introduction, lines 78-83.

Comment 3: Figure 2 should be representative of the entire groups of patients not only selected results for endometriosis and for ovarian cancer. It should be shown as a box plot to depict the distribution of all individual values for control and a box plot for all the O-ASC CM-exposed. One panel should represent the endometriosis cells treated with autologous O-ASC CM, and another box plot should represent endometriosis cells treated with ovarian cancer O-ASC CM. Further, in the second panel, it should present separate box plots with cases of HGSC,CCC, EC and MC cells treated with respective O-ASC CM.
Response 3: Thank you for the suggestion. We have modified Figure 2 to box plots that show individual values. We have also modified the figure legend and panel labels in the main text accordingly, lines 127-177.

Comment 4: Line 172 do you mean matrix metalloproteinase instead of metallopeptidase ?
Response 4: Thank you for pointing this out. We have corrected matrix metallopeptidase to matrix metalloproteinase in line 199.

Comment 5: Characterizing the secretome of omental-adipose stromal cells from the 29 samples of O-ASC CM is problematic due to numbers of samples of subtypes: malignant (11 HGSC, 8 CCC, 3 EC, 1 MC), to borderline tumors (2 BorMT), endometriosis (3 EMS) and benign cyst (1 BeCy). The types where you have at least 3 individual samples could be retained in the analysis, but having only 1 for MC and BeCy and 2 for BorMT is not enough for these analyses. I would recommend excluding them from these analyses. Also, cytokines were consistently more highly secreted in HGSC. Since you have a much greater sample size for HGSC compared to other subtypes, this could influence the analysis.
Response 5: Thank you for the suggestion. We have excluded MC, BorMT and BeCy from Figure 3. We have updated the results section and figure legend accordingly, lines 222-237. We retained the original data for MC, BorMT and BeCy in Supplementary Table S4 and Figure S3.

Comment 6: In line 202 you state that cells were each exposed to histologically matched O-ASC CM but that is not correct since in the bracket you state that a single sample E377 was used for the treatment of all EMS and EC cells. Also, a single KO20 O-ASC CM was used for the treatment of all HGSC cells from 4 donors. Thus, you have only 2 different conditioned media for the treatment of all cell samples.
Response 6: Thank you for the comment. The patient ID for O-ASC CM are very similar and might have been mistaken as the same. E377 O-ASC CM was used for EMS cells, E337 O-ASC CM was used for EC cells and KO20 was used for HGSC cells. Therefore there are 3 different O-ASC CM, one for each histological subtype.

Comment 7: To determine the differences in HGF/c-MET signaling in Control, O-ASC CM treated group and inhibitor+ O-ASC CM group, ANOVA should be used instead of t-test.
Response 7: Thank you for the suggestion. For Figure 5, we changed the statistical test to ANOVA followed by Tukey’s multiple comparison test to determine the statistical significance between each group. The figure legend has been updated in lines 297-298 and p-values are also updated in Supplementary Table S2.

Comment 8: Line 266 instead of cell proliferation MTT rather analyses cell viability, as it represents the number of metabolically active cells. Include the MTT results from supplementary in the main manuscript as you describe these results in the section 266-271. Also, add explanation why you evaluated cell viability for 6 days.
Response 8: Thank you for the suggestion. We have moved the MTT results from Supplementary to main manuscript, Figure 6. We also provided further explanation on MTT results, in lines 322-327.

Comment 9: Line 278 “primary cultures of endometriosis, ovarian cancer” correct to “primary cultures established from the tissue samples of endometriosis and ovarian cancer patients”
Response 9: Thank you for the suggestion. We have updated the sentence in the discussion, lines 357-358.

Comment 10: Line 278-283 should be rephrased, it is not clear what the authors meant in these parts: “maximally reflect their usual functional states,” and “tissue’s microcosm of cell subpopulations in vivo, which may positively influence their longevity ex vivo”
Response 10: Thank you for the comment. We have rephrased the statements in the Discussion, lines 358-362.

Comment 11: Line 317 there is a reference no 1 in superscript at the end of the sentence. Discussion should be improved by citing some other findings relevant to be compared with your results.
Response 11: Thank you for pointing this out. We have removed (1) and added in the correct reference (ref 39) at line 423.

Comment 12: Lines 318-321 are generalized statements that do not provide any clear conclusion.
Response 12: Thank you for the comment. We have removed these generalized statements from the main manuscript.

Comment 13: Line 324” Notably, the O-ASC CM from the patient with BeCy did not exhibit angiogenic potential.” Since you had only a single sample of BeCy O-ASC CM it is very likely that by increasing sample size, some differences could be observed. Add here the limitations of this study by referring to number of samples.
Response 13: Thank you for the comment. We agree that the BeCy O-ASC CM sample size is too small. We have included the limitation on the single data point in line 429.

Comment 14: Line 342 “inhibition of PTTG1 may delay the progression or recurrence of endometriosis and ovarian cancer” is there a reference for this statement or this is an author’s assumption? If it is an assumption, add explanations, since you did not evaluate the effects of PPTG1 expression inhibition.
Response 14: Thank you for the comment. We have rephrased the statements and included references in lines 444-448.

Comment 15: Improve the explanation of the limitations of this study in the final paragraph, as they are poorly written.
Response 15: Thank you for the comment. We have rewritten the final paragraph, to better explain the limitations of the study, in lines 450-508.

Reviewer 3 Report

Comments and Suggestions for Authors

It looks like a thorough study with lots of interesting data and samples from human patients with clinical relevance.

Can you comment on the pros and cons of using trans well migration assay and how it relates to your experiment.

Can you comment on how you imagine using alternative approaches to the trans well assays.

Can you do repeated trans well migration assays? For example collect the cells that pass through, wash them and make them migrate across again. Do you think there is a heterogeneous response? I do not require you do this just trying to think of ways to get information about cancer cell heterogeneity.

When you pool the cells into nanostring you average over the population cells. They are heterogeneous in the sense of those that pass and those that don't. It is not 100% that they pass so there is heterogeneity. Are there any platforms that analyze omics of single cells?

A lot of the data is very busy with large numbers and many digits. Can you implement error bars. For example if you measure 100 cells then the error is ten. You might want to use significant digits better. There is a range of data but you show to the hundredths with five significant digits, when in fact you only measure hundreds or so. The error is 10 percent or a few percent. For example you would report 100(10). Or another measurement might be 4.5(3) which is 4.5 pm 0.3.

Author Response

Response to Comments from Reviewer 3

Thank you very much for taking the time to review this manuscript. Please find the detailed responses in blue below and the corresponding revisions are highlighted/in track changes in the resubmitted files.

Comment 1: Can you comment on the pros and cons of using trans well migration assay and how it relates to your experiment.
Response 1: Thank you for the comment. The transwell migration assay is simple and easy to setup, and the number of migrated cells can be easily quantified by staining and ImageJ analysis. We used this assay to study how the omentum and its microenvironment (through the use of omentum conditioned media) influences the migration of endometriosis or ovarian cancer cells.

The main disadvantage of transwell migration assay is that the setup is two-dimensional and non-physiological, which does not accurately reflect the endogenous 3D microenvironment in the abdomen-pelvic cavity. The assay may also miss out other cell-cell interactions present in the 3D microenvironment, that could potentially affect the migration dynamics. Additionally, the transwell migration assay is an endpoint assay, which does not capture cell migration in real-time.

We have added the limitations of the transwell migration assay into the manuscript under Discussion, lines 450-457.    

Comment 2: Can you comment on how you imagine using alternative approaches to the trans well assays.
Response 2: Thank you for the comment. Alternatives to transwell migration assays include (i) scratch/wound-healing assay, (ii) xCELLigence assay, (iii) microfluidic devices in combination with time-lapse imaging to capture the migration dynamics in real-time and to mimic physiological condition. In addition to the transwell migration assay, we have performed the scratch/wound-healing assay (Supplementary Figure S2) and obtained results that were consistent with the transwell migration assay.

Comment 3: Can you do repeated trans well migration assays? For example collect the cells that pass through, wash them and make them migrate across again. Do you think there is a heterogeneous response? I do not require you do this just trying to think of ways to get information about cancer cell heterogeneity.
Response 3: Thank you for the comment. For the migrated cells (i.e. cells that passed through to the underside of the membrane), it will be difficult to flush them out of the membrane and perform another round of transwell migration. The membrane has to be cut open to retrieve the cells on the underside of the membrane and in doing so, the cells are unlikely to survive. Therefore, it is not possible to perform repeated transwell migration assay.

Comment 4: When you pool the cells into nanostring you average over the population cells. They are heterogeneous in the sense of those that pass and those that don't. It is not 100% that they pass so there is heterogeneity. Are there any platforms that analyze omics of single cells?
Response 4: Thank you for the comment. To clarify, we extracted the RNA of migrated cells (i.e. cells that passed through to the underside of the membrane; the red cells in Figure 4A) and RNA of unmigrated cells (i.e. cells that were washed out from the transwell membrane; the blue cells in Figure 4A). We agree that single cell analysis such as single cell RNA sequencing will more accurately decipher the heterogeneity of cells compared to bulk RNA analysis (e.g. Nanostring). The cost of single cell analysis is about 10 times higher than bulk analysis and we did not pursue the single cell analysis due to budget constraint. 

Comment 5: A lot of the data is very busy with large numbers and many digits. Can you implement error bars. For example if you measure 100 cells then the error is ten. You might want to use significant digits better. There is a range of data but you show to the hundredths with five significant digits, when in fact you only measure hundreds or so. The error is 10 percent or a few percent. For example you would report 100(10). Or another measurement might be 4.5(3) which is 4.5 pm 0.3.
Response 5: Thank you for the comment. Error bars are presented in Figures 2, 3, 5, 6 to show the range of statistical variability. 

Round 2

Reviewer 2 Report

Comments and Suggestions for Authors

The Authors have adressed most of the concerns in the revised version of the manuscript. However, there are few smaller issues to be corrected prior to publication:

Represent Fig3 as box plots, rather than bars, since you have a distribution of the values. How did you observe significant difference for HGF in panel A between HGSC and CCC having in mind the standard deviation of HGCS? Check again this difference. Also, in each figure with results add in the figure legend the type of statistics used (e.g. unpaired student’s t-test, Mann Whitney U test or ANOVA followed by Tukey’s multiple comparison test) and Instead of ±  put sign + since in all the graphs and box plots you depicted this representation of SD only on the upper side of bars.

Check the significance of the differences in Fig 5. in panel F,  you can not have significant difference with these standard deviations presented.

In line 352, add also  MC, since it also had single data point , and  BorMT since it had 2 values. Again, anything less than 3 can not be analyzed properly.
